# Safety of Janssen Ad26.COV.S and Astra Zeneca AZD1222 COVID-19 Vaccines among Mobile Phone Users in Malawi: Findings from a National Mobile-Based Syndromic Surveillance Survey, July 2021 to December 2021

**DOI:** 10.3390/ijerph20237123

**Published:** 2023-11-30

**Authors:** Lucky Makonokaya, Lester Kapanda, Godfrey B. Woelk, Annie Chauma-Mwale, Louiser Upile Kalitera, Harrid Nkhoma, Suzgo Zimba, Rachel Chamanga, Cathy Golowa, Rhoderick Machekano, Thulani Maphosa

**Affiliations:** 1Elizabeth Glaser Pediatric AIDS Foundation, Lilongwe P/Bag 2543, Malawirkanyenda@pedaids.org (R.C.); cgolowa@pedaids.org (C.G.); tmaphosa@pedaids.org (T.M.); 2Elizabeth Glaser Pediatric AIDS Foundation, Washington, DC 20005, USA; gwoelk@pedaids.org (G.B.W.); rmachekano@pedaids.org (R.M.); 3Public Health Institute, Ministry of Health Malawi, Lilongwe P.O. Box 30377, Malawi

**Keywords:** COVID-19, SARS-CoV-2, adverse events, Ad26.COV2.S, AZD1222, Malawi, mobile phone-based syndromic surveillance survey, vaccination, self-report

## Abstract

The safety profiles of the Ad26.COV2.S and AZD1222 COVID-19 vaccines have not been described in the general population in Malawi. We present self-reported adverse events (AE) following the receipt of these vaccines in Malawi as part of a national syndromic surveillance survey. We conducted phone-based syndromic surveillance surveys among adults (≥18 years) with verbal consent. We used secure tablets through random digit dialing to select mobile phone numbers and collected data electronically. Survey questions included whether the respondent had received the COVID-19 vaccines, whether they had experienced any AE following vaccination, and the severity of the AE. We used multivariable analysis to identify factors associated with self-reported AE post-COVID-19 vaccination. A total of 11,924 (36.0%) out of 33,150 respondents reported receiving at least one dose of either Ad26.COV2.S or AZD1222 between July–December 2021; of those, 65.1% were female. About 49.2% of the vaccine recipients reported at least one AE, 90.6% of which were mild, and 2.6% were severe. Higher education level and concern about the safety of COVID-19 vaccines were associated with AE self-report (Adjusted Odds Ratio [AOR] 2.63 [95% CI 1.96–3.53] and 1.44, [95% CI 1.30–1.61], respectively), while male gender and older age were associated with reduced likelihood of AE self-report (AORs 0.81, [95% CI 0.75–0.88], 0.62 [95% CI 0.50–0.77], respectively). Ad26.COV2.S and AZD1222 vaccines are well-tolerated, with primarily mild and few severe AE among adults living in Malawi. Self-reporting of AE following COVID-19 vaccination is associated with gender, age, education, and concern about the safety of the vaccines. Recognizing these associations is key when designing and implementing COVID-19 vaccination communication messages to increase vaccination coverage.

## 1. Introduction

The novel Coronavirus Disease 2019 (COVID-19) caused by Severe Acute Respiratory Syndrome Coronavirus 2 (SARS-CoV-2) infection led to a remarkable disease burden globally, resulting in over 767 million registered cases and 6.9 million deaths associated with the pandemic [1]. While infection prevention and control measures such as mask-wearing in public places, hand washing, and sanitizing helped reduce the spread of the infection and resultant disease, the implementation of these measures was a challenge globally, especially in Low- and Middle-Income Countries (LMICs) [2,3]. Vaccines reduce morbidity and mortality due to COVID-19 [4,5]. In the absence of a cure, vaccines are vital in the fight against the COVID-19 pandemic [6,7]. As of 7 July 2022, the World Health Organization (WHO) had recommended ten vaccines against COVID-19 [8].

Over 12.2 billion vaccine doses have been administered globally, with 4.8 billion people fully vaccinated against COVID-19 [9]. However, there has been slow uptake of vaccines in LMIC compared to High-Income Countries (HIC) [9]. Initial lack of availability and, more recently, concern about the safety of vaccines, among other factors, have contributed to low vaccine uptake in LMICs [10,11].

Ad26.COV2.S and AZD1222 vaccines were among the early COVID-19 vaccines recommended by the WHO [12,13]. Ad26.COV2.S, an adenovirus-vectored vaccine given as a single dose, protects against moderate to severe COVID-19 with an efficacy of 66.9% [14]. AZD1222, another adenovirus-vectored vaccine, is administered in two doses with an efficacy of 74.0% [15]. Vaccine trials and early implementation studies demonstrated that these vaccines are safe, with most of the local and systemic AE reported being mild to moderate [14,15,16]. The most commonly reported AE include myalgia, fatigue, headache, pain on the injection site, and fever. Potentially life-threatening AE, such as anaphylaxis, thromboembolic events, and Guillain–Barré syndrome, are rarely associated with Ad26.COV2.S and AZD1222 vaccines [14,15,16].

The estimated population of Malawi is 17.5 million, with more than half aged below 18 years [17]. The country rolled out the COVID-19 vaccination program with the AZD1222 vaccine in March 2021, followed by the Ad26.COV2.S vaccine in September 2021 [18]. Both of these vaccines were offered free to all residents of the country who were at least 12 years old. However, based on availability, some people did not have the opportunity to choose between the two vaccines. The nation intended to vaccinate at least 60% of the population by the end of 2022. Nevertheless, despite multiple efforts by the government and other implementing partners, only about 7% of the nation’s population had been vaccinated against COVID-19 at the beginning of August 2022 [9].

In a study conducted among healthcare workers in Malawi, 71.2% of COVID-19 vaccine recipients reported mild to moderate adverse events (AE). Fear of AE was the most common reason for declining COVID-19 vaccination [19]. There remains a paucity of data on population-based studies in the region looking at vaccine safety and a lack of a national reporting system of AE following COVID-19 vaccine receipt. Assessing the tolerability of these vaccines in the local setting would help provide a better context for messages on the safety of the vaccines.

The COVID-19 syndromic surveillance study provided an opportunity to evaluate the safety of COVID-19 vaccines in a large, diverse population. This survey, involving the random calling of mobile phone numbers throughout Malawi, was initiated as a low-cost surveillance strategy to monitor the progression of the COVID-19 epidemic, given the limitations of traditional surveillance methods. We subsequently sought to include questions on vaccinations, including reported vaccine adverse events among vaccine recipients of the two vaccines available in Malawi.

## 2. Methods

### 2.1. Study Design

Between July 2020 and April 2022, the Elizabeth Glaser Pediatric AIDS Foundation (EGPAF) in Malawi, in collaboration with the Public Health Institute of Malawi (PHIM), conducted a mobile phone-based cross-sectional COVID-19 syndromic surveillance survey among adults living in Malawi with access to an active mobile phone number. However, we utilized data on COVID-19 vaccination collected between July 2021 to December 2021. Survey questions included age, gender, education, region of residence, getting vaccination if recommended, concerns about severe reactions to the vaccines, family decision-making on the vaccination, the influence of close friends and family on vaccine receipt, trust in the vaccines, thoughts on the safety of the vaccines, vaccination, number of doses received, adverse events experienced following vaccine receipt, the severity of adverse events experienced, seeking medical care due to the adverse events, and hospitalization due to the adverse events.

To gather data for our research project, we made phone calls to computer-generated random numbers belonging to the country’s two national mobile phone network providers. We identified research assistants with a background in healthcare to conduct phone-based interviews and enter the data in real-time using tablets. The research assistants were trained in conducting phone-based surveys, the study protocol, and research ethics for human subjects.

The interviews were conducted via mobile phone, and participants were given the option to choose between English or Chichewa as their preferred language for the interview. We ensured that all respondents were at least 18 years old and obtained their verbal consent before they could participate in the survey. We needed to include participants who could effectively communicate in either English or Chichewa, so individuals who could not do so were excluded from participating.

To facilitate the survey process, we developed a questionnaire in English, which was then translated into Chichewa. To ensure that the translated version retained the same meaning as the original English version, we conducted a back-translation of the questionnaire. The questionnaire was then digitized, incorporating data range and consistency checks to maintain high data quality. We conducted a pilot of the questionnaire before implementing it.

We used Android tablets with Open Data Kit (ODK) collect v1.27 software for data collection, which allowed us to capture the survey responses electronically, ensuring efficient and accurate data collection. The collected data were securely stored on an online server to maintain confidentiality and accessibility.

### 2.2. Sample Size

Sample size estimates were based on the primary objective of estimating the weekly rates of self-reported influenza-like or COVID-19-like illnesses (ILI/CLI). Without preliminary data on rates of ILI/CLI in Malawi, we conservatively assumed that 50% of the population had symptoms of ILI/CLI. We estimated that 1537 individuals would allow us to estimate ILI/CLI weekly rates with 2.5% precision. We planned to survey at least 2000 individuals from the general population per week. This sample size was estimated for the original survey’s primary objective. For the vaccination studies, the sample size was considered sufficient for a precision estimate of +/−5% for the commonly reported side effect of pain at the injection site [14,15,16].

### 2.3. Study Variables

We asked the respondents whether they had received any COVID-19 vaccines and the type of vaccine received. For this analysis, we focused on examining the self-reported adverse events that individuals experienced after receiving COVID-19 vaccines. We asked respondents whether they had any AE following their most recent dose of COVID-19 vaccines. Those who reported AE were further asked to grade the AE on a scale of mild, moderate, or severe based on their experience. This served as the outcome variable, allowing us to investigate the potential adverse events associated with the vaccination process. We considered several independent variables to explore the factors that could influence adverse events. Sociodemographic characteristics played a significant role, including gender, age, current region of residence, and the highest level of education attained. By analyzing these factors, we aimed to understand whether certain demographic groups were more susceptible to adverse events or if there were any patterns based on these characteristics. Additionally, we sought to explore participants’ concerns regarding contracting the COVID-19 disease. Another important independent variable was the participants’ thoughts on the safety of COVID-19 vaccines. Furthermore, we considered the participants’ final say on vaccine receipt, which reflected their decision-making authority regarding vaccination. By understanding the extent of individuals’ autonomy in deciding whether to receive the vaccine, we could evaluate if this factor had any impact on the occurrence of adverse events. By examining these independent variables, we aimed to understand the factors influencing the self-reporting of adverse events following COVID-19 vaccine receipt. This analysis would provide valuable insights into the vaccine’s safety profile and help inform future vaccination campaigns and policies.

### 2.4. Statistical Analysis

Data analysis was performed using STATA v16. Frequencies and proportions were used to summarize the distribution of categorical demographic characteristics of COVID-19 vaccine recipients and the characteristics of respondents who reported having adverse events among vaccine recipients. Medians and interquartile ranges were used to summarize continuous variables such as age.

Pearson’s Chi-square test was used to test for potential associations between self-reported adverse events and various covariates. We used multivariable logistic regression to determine the factors independently associated with reported adverse reactions among vaccine recipients, adjusting for gender, age, current region of residence, the highest level of education attained, concern about getting COVID-19 disease, thoughts on the safety of COVID-19 vaccines, and final say on COVID-19 vaccine receipt. All the independent variables were included in the multivariate model.

### 2.5. Ethical Considerations

The Malawi National Health Sciences Research Committee (protocol number 22/06/2537) and Advarra Institutional Review Board (protocol number Pro00045270) in the United States of America (USA) approved the study. We obtained verbal informed consent from all study participants.

## 3. Results

We made 189,425 calls between July and December 2021; 67.3% (n = 127,511) were active numbers. Nearly 39.4% (n = 50,235) of the active numbers were answered; 67.1% (n = 33,688) were eligible for study participation. All of the eligible respondents consented to study participation, 98.2% (n = 33,080) of whom completed interviews (Figure 1).

A total of 11,924 (36.0%) respondents reported receipt of at least one dose of COVID-19 vaccines. A total of 3891 respondents (32.6%) had received one dose of the AZD1222 vaccine. In contrast, 5623 (47.2%) had received two doses of the AZD1222 vaccine, and 2409 (20.2%) respondents had received the Ad26.COV2.S vaccine (Figure 2).

Most of the vaccine recipients (65.1%) were female. The median age was 38 years, and the interquartile range (IQR) was 29–47 years. Approximately 42% of vaccine recipients had attained secondary education, and 44.6% resided in the central region (Table 1).

Overall, 49.2% of the respondents who received the vaccines reported at least one AE following COVID-19 vaccine receipt. There were significant differences in self-reporting of AE among different sociodemographic groups: more females (52.6%) than males (47.4%), more adults aged 35–44 years (52.8%) than those aged 18–24 (45.5%), and more who had attained tertiary education (60.4%) than those with no formal education (36.1%). The incidence of AE in respondents who received Ad26.COV2.S vaccine was 99% (n = 2385) versus 16.9% (n = 656) in those who received the first dose of AZD1222 and 50.2% (n = 2823) in those who received the second dose of AZD1222 (Table 2).

The most commonly reported AE were joint pain (45.5%), fever (26.7%), headache (26.1%), pain at the injection site (24.4%), and fatigue (16.6%). Other reported AE (>1%) included chills (15.7%), dizziness (7.6%), nausea (3.9%), and wheezing (1.4%). Approximately 90.6% of the reported AE were mild, 6.8% were moderate, and 2.6% were severe. There were no significant differences in those who sought medical care or were hospitalized by gender or age. A total of 320 (5.6%) participants were treated as outpatients following AE, 41 (0.7%) reported being hospitalized, and 5408 (93.7%) did not seek any medical attention following the AE (Table 3).

In a multivariate analysis, males were less likely to report post-vaccination AE compared to females (Adjusted Odds Ratio (AOR) 0.81, 95% Confidence Interval (CI) 0.75–0.88). Respondents aged 25–34 years (AOR = 1.24, 95% CI 1.09–1.41) and 35–44 years (AOR = 1.31, 95% CI 1.15–1.50) had increased odds of reported AE following COVID-19 vaccine receipt compared to those aged 18–24 years. However, respondents aged 65 years or older had lower odds of reporting AE than 18–24-year-olds (AOR = 0.62, 95% CI 0.50–0.77). The odds of post-vaccination AE were higher in participants with secondary (AOR = 1.62, 95% CI 1.22–2.18) and tertiary education (AOR = 2.63 (95% CI 1.96–3.53) compared to those with no formal education. Respondents who were residing in the southern region had increased odds of reporting AE post-COVID-19 vaccination than those residing in the northern region. Respondents who thought COVID-19 vaccines were unsafe were more likely to report AE (AOR = 1.44, 95% CI 1.27–1.64) compared to those who believed the vaccines were safe (Table 4).

## 4. Discussion

Our study showed that the Ad26.COV2.S and AZD1222 vaccines are generally well-tolerated among the adult population with phone ownership in Malawi. Nearly half of the respondents who reported receipt of at least one dose of the COVID-19 vaccines had AE. There were no significant differences in the most commonly reported AE between those who received the first and second doses of AZD1222 and those who received Ad26.COV2.S. Most respondents had mild AE following the COVID-19 vaccine receipt and never sought medical treatment following the adverse events. Joint pain, fever, headache, pain at the injection site, and fatigue were the most commonly reported post-vaccination AE. These findings are similar to those of phase 3 trials and other studies conducted in the region [15,16,20]. Potentially life-threatening AE, including anaphylaxis, thromboembolic events, and Guillain–Barré syndrome, have been reported in the phase 3 trials and in other studies conducted in the sub-Saharan region [15,16,21]. However, these AE were not reported in our study. Patients who experience such AE are more likely to seek medical attention, and improved post-vaccination AE reporting systems are needed in health facilities across the country to help monitor these events.

Self-reports of post-vaccination AE varied across demographic characteristics. Females were more likely to report AE compared to males, which was similar to other studies [15,20,22]. Females have demonstrated more robust immune systems and greater immune responses following vaccine receipt than males at all stages of life [23,24]. There may have been underreporting of AE from the males due to the poor health-seeking behavior demonstrated by males in this region [25]; however, the scope of this study could not evaluate this characteristic. The possible underreporting of adverse events among males could be due also to gender role expectations, where males are expected to be “strong” and to feel less pain and discomfort in the face of illness [25,26].

Similar to other studies, older age was associated with a reduced likelihood of post-vaccination AE [20,27]. These findings were expected because older people have weaker immune systems, leading to lower antibody titers and lower reactogenicity rates than younger people. Self-report of AE also increased with an increased level of education. We hypothesize this may be due to increased awareness of AE because those who are more educated are more likely to know about AE and subsequently more likely to report the symptom as an AE. It may also be due to better overall health-seeking behavior demonstrated by those who are more educated when compared to those who are less educated [28,29,30].

More respondents who received the second dose of AZD1222 reported post-vaccination AE compared to the first dose, similar to findings from other studies [31,32]. This pattern is observed often because the first dose serves as a primer to the immune system, and the second dose serves as a booster to enhance and further strengthen the immune response. It is also worth noting that only about one-fifth of the respondents who received one dose of AZD1222 reported adverse events, which was significantly lower than in other studies. Our findings may be partly explained by the respondents’ increased awareness of the vaccine and its side effects, which would have been better among those receiving the second dose than the first one. There may also have been differences in durations between the first and second doses, leading to differences in vaccine receipt immunogenicity [33,34]; however, this factor was not measured in our study.

The likelihood of reporting post-vaccination AE was also associated with the perception of the safety of the vaccines. Respondents who thought vaccines were unsafe were more likely to report adverse events than those who thought they were safe. We hypothesize that the participants’ thoughts of the vaccine’s safety may have been influenced by their experience of post-vaccination adverse events, which would give perceptions of vaccines being unsafe if they had more adverse events and vice versa.

We recognize the limitations of our study. First, the study was based on self-reporting, with no other data sources to triangulate with, predisposing it to recall and social desirability bias. Second, the distribution of the respondents’ demographic characteristics mirrored that of the country’s mobile phone ownership, with more males than females and most respondents aged between 25 and 44 years [35], compared to the country’s population profile of 51% female, and about half under 18 years [17]. Also, our study respondents were more likely to reside in urban areas because mobile phone ownership is higher in urban areas than in rural areas [35]. Hence, the findings need to be generalized with caution. However, due to random sampling and the large sample size, different demographic characteristics were well-represented.

## 5. Conclusions

Fewer AE were reported among an adult population living in Malawi following receipt of Ad26.COV2.S and AZD1222 vaccines than reported in phase 3 trials. Female gender, younger age, higher level of education, and concern about the safety of the vaccines were associated with an increased likelihood of post-vaccination adverse events. Recognizing these associations is critical when designing and implementing COVID-19 vaccination communication messages to increase vaccination coverage.

## Figures and Tables

**Figure 1 ijerph-20-07123-f001:**
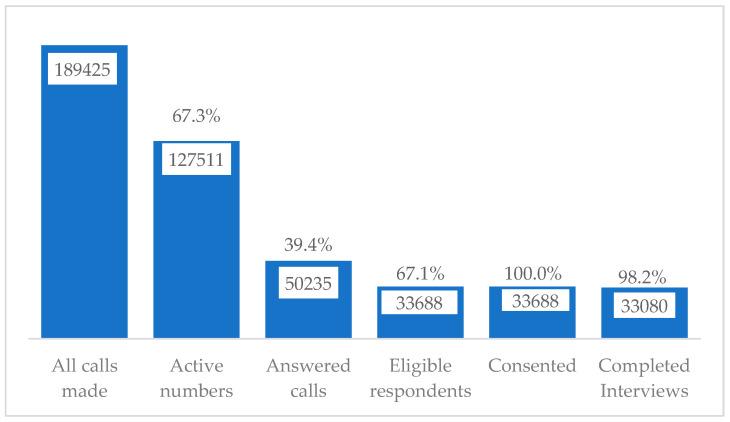
Enrollment cascade of the COVID-19 syndromic surveillance study respondents, July 2021 to December 2021.

**Figure 2 ijerph-20-07123-f002:**
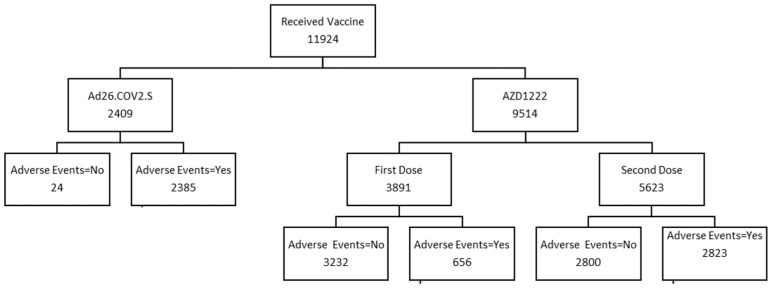
Flow of participants who self-reported COVID-19 vaccine receipt during syndromic surveillance survey between July 2021 and December 2021.

**Table 1 ijerph-20-07123-t001:** Distribution of demographic characteristics of COVID-19 vaccine recipients between July and December 2021 (n = 11,924).

Variable	Frequency n (%)
Gender	
Female	4165 (34.9%)
Male	7759 (65.1%)
Age range	
	(Median Age: 38, IQR: 29, 47)
18–24	1493 (12.5%)
25–34	3263 (27.4%)
35–44	3298 (27.7%)
45–54	2340 (19.6%)
55–64	993 (8.3%)
65+	537 (4.5%)
Level of Education	
No Education	209 (1.8%)
Primary	3330 (28.1%)
Secondary	4955 (41.7%)
Tertiary	3378 (28.5%)
missing	52
Region of Residence	
Northern	1943 (16.3%)
Central	5323 (44.6%)
Southern	4658 (39.1%)
Type of Vaccine	
AZD1222 1st Dose	3891 (32.6%)
AZD1222 2nd Dose	5623 (47.2%)
Ad26.COV2.S	2409 (20.2%)
missing	1
Concern about getting COVID-19 disease	
Little Concerned	2548 (21.4%)
Very Concerned	9376 (78.6%)
Thoughts on the Safety of COVID-19 Vaccines	
Safe	10,218 (85.7%)
Not safe	1706 (14.3%)
Final say on Vaccine receipt	
Self	10,469 (87.8%)
Spouse	909 (7.6%)
Parents/In-laws	359 (3%)
Children	62 (0.5%)
Someone else	125 (1.1%)

**Table 2 ijerph-20-07123-t002:** Characteristics of respondents who reported adverse events among those who received the COVID-19 vaccine.

Variable	Frequency n (%)	*p*-Value
Gender		
Female	2188 (52.6%)	<0.01
Male	3676 (47.4%)	
Age range		
18–24	678 (45.5%)	<0.01
25–34	1719 (52.7%)	
35–44	1743 (52.8%)	
45–54	1117 (47.8%)	
55–64	430 (43.3%)	
65+	177 (33%)	
Level of Education		
No Education	75 (36.1%)	<0.01
Primary	1338 (40.2%)	
Secondary	2390 (48.2%)	
Tertiary	2041 (60.4%)	
Region of Residence		
Northern	881 (45.3%)	<0.01
Central	2659 (50.0%)	
Southern	2324 (49.9%)	
Type of Vaccine		
AZD1222 1st Dose	656 (16.9%)	<0.01
AZD1222 2nd Dose	2823 (50.2%)	
Ad26.COV2.S	2385 (99.0%)	
Concern about getting COVID-19 disease		
Little Concerned	1211 (47.5%)	0.06
Very Concerned	4653 (49.7%)	
Thoughts on the Safety of COVID-19 Vaccines		
Safe	4875 (47.7%)	<0.01
Not safe	989 (58.0%)	
Final say on Vaccine receipt		
Self	5088 (48.6%)	0.01
Spouse	494 (54.4%)	
Parents/In-laws	182 (50.7%)	
Children	33 (53.2%)	
Someone else	67 (53.6%)	

**Table 3 ijerph-20-07123-t003:** Adverse events after COVID-19 vaccine receipt, the severity of adverse events, medical treatment, and hospitalizations according to gender and age groups.

Variables	Total	Gender		Age Range					
		Female (n = 2188)	Male (n = 3676)	18–24 (n = 678)	25–34 (n = 1719)	35–44 (n = 1743)	45–54 (n = 1117)	55–64 (n = 430)	65+ (n = 177)
Adverse Events									
Joint pain	2668 (45.5%)	1012 (46.3%)	1656 (45.1%)	303 (44.7%)	802 (46.7%)	815 (46.8%)	501 (44.9%)	184 (42.8%)	63 (35.6%)
Fever	1566 (26.7%)	569 (26.0%)	997 (27.1%)	162 (23.9%)	497 (28.9%)	479 (27.5%)	287 (25.7%)	100 (23.3%)	41 (23.2%)
Headache	1529 (26.1%)	650 (29.7%)	879 (23.9%)	177 (26.1%)	456 (26.5%)	457 (26.2%)	290 (26.0%)	112 (26.1%)	37 (20.9%)
Pain at the injection site	1430 (24.4%)	550 (25.1%)	880 (23.9%)	187 (27.6%)	375 (21.8%)	432 (24.8%)	290 (26.0%)	103 (24.0%)	43 (24.3%)
Fatigue	975 (16.6%)	312 (14.3%)	663 (18.0%)	152 (22.4%)	323 (18.8%)	256 (14.7%)	162 (14.5%)	59 (13.7%)	23 (13.0%)
Chills	918 (15.7%)	378 (17.3%)	540 (14.7%)	112 (16.5%)	300 (17.5%)	265 (15.2%)	176 (15.8%)	47 (10.9%)	18 (10.2%)
Dizziness	446 (7.6%)	194 (8.9%)	252 (6.9%)	66 (9.7%)	147 (8.6%)	116 (6.7%)	83 (7.4%)	25 (5.8%)	9 (5.1%)
Nausea	227 (3.9%)	122 (5.6%)	105 (2.9%)	27 (4.0%)	58 (3.4%)	75 (4.3%)	47 (4.2%)	14 (3.3%)	6 (3.4%)
wheezing	82 (1.4%)	34 (1.6%)	48 (1.3%)	21 (3.1%)	25 (1.5%)	17 (1.0%)	13 (1.2%)	2 (0.5%)	4 (2.3%)
Other	367 (6.3%)	155 (7.1%)	212 (5.8%)	41 (6.1%)	102 (5.9%)	105 (6.0%)	77 (6.9%)	28 (6.5%)	14 (7.9%)
The Severity of Adverse Events									
Mild	5229 (90.6%)	1923 (88.9%)	3306 (91.6%)	585 (88.5%)	1496 (88.5%)	1569 (91.1%)	1013 (91.9%)	399 (94.1%)	167 (96.5%)
Moderate	397 (6.8%)	160 (7.4%)	237 (6.6%)	46 (7.0%)	148 (8.8%)	120 (7.0%)	60 (5.4%)	19 (4.5%)	4 (2.3%)
Severe	147 (2.6%)	81 (3.7%)	66 (1.8%)	30 (4.5%)	47 (2.8%)	33 (1.9%)	29 (2.6%)	6 (1.4%)	2 (1.2%)
Treated as outpatients	320 (5.6%)	125 (5.8%)	195 (5.4%)	45 (6.8%)	98 (5.8%)	96 (5.6%)	59 (5.3%)	15 (3.5%)	7 (4.0%)
Hospitalized	41 (0.7%)	18 (0.8%)	23 (0.6%)	9 (1.4%)	13 (0.8%)	11 (0.6%)	4 (0.4%)	2 (0.5%)	2 (1.2%)
Did not seek any health care service	5408 (93.7%)	2020 (93.4%)	3606 (94%)	607 (91.8%)	1580 (93.4%)	1612 (93.8%)	1038 (94.3%)	407 (96%)	164 (94.8%)

Data presented as number (%).

**Table 4 ijerph-20-07123-t004:** Factors associated with reported adverse events among COVID-19 vaccine recipients (Logistic regression).

Variable	Adverse Event = Yes n (%)	Unadjusted Odds Ratio (95% CI)	*p* Value	Adjusted Odds Ratio (95% CI)	*p* Value
Gender					
Female	2188 (52.3%)	Ref		1	
Male	3676 (47.4%)	0.81 (0.75–0.88)	<0.01	0.81 (0.75–0.88)	<0.01
Age range					
18–24	678 (45.5%)	Ref		1	
25–34	1719 (52.68%)	1.34 (1.18–1.51)	<0.01	1.24 (1.09–1.41)	<0.01
35–44	1743 (52.9%)	1.34 (1.19–1.52)	<0.01	1.31 (1.15–1.50)	<0.01
45–54	1117 (47.8%)	1.1 (0.96–1.25)	0.17	1.12 (0.98–1.29)	0.09
55–64	430 (43.3%)	0.92 (0.78–1.08)	0.29	0.97 (0.82–1.15)	0.75
65+	177 (33%)	0.59 (0.48–0.72)	<0.01	0.62 (0.50–0.77)	<0.01
Level of Education					
No Education	75 (36.1%)	Ref		1	
Primary	1338 (40.2%)	1.19 (0.89–1.59)	0.24	1.19 (0.88–1.60)	0.25
Secondary	2390 (48.2%)	1.65 (1.24–2.2)	<0.01	1.62 (1.22–2.18)	<0.01
Tertiary	2041 (60.4%)	2.71 (2.02–3.62)	<0.01	2.63 (1.96–3.53)	<0.01
Region					
Northern	881 (45.3%)	Ref		1	
Central	2659 (50%)	1.20 (1.08–1.34	<0.01	1.13 (0.99–1.29)	0.07
Southern	2324 (49.9%)	1.20 (1.08–1.34)	<0.01	1.12 (1.05–1.37)	<0.01
Concern about getting COVID-19 disease					
Little Concerned	1211 (47.5%)	Ref		1	
Very Concerned	4653 (49.7%)	1.09 (1.36–1.68)	0.06	1.06 (0.97–1.17)	0.17
Thoughts on the Safety of COVID-19 Vaccines					
Safe	4875 (47.7%)	Ref		1	
Not safe	989 (58%)	0.91 (0.88–0.95)	<0.01	1.44 (1.30–1.61)	<0.01
Final say on Vaccine receipt					
Self	5088 (48.6%)	Ref		1	
Spouse	494 (54.4%)	1.26 (1.10–1.44)	<0.01	1.26 (1.10–1.45)	<0.01
Parents/In-laws	182 (50.7%)	1.08 (0.88–1.34)	0.44	1.14 (0.91–1.42)	0.25
Children	33 (53.2%)	1.20 (0.73–1.98)	0.47	1.39 (0.84–1.74)	0.3
Someone else	67 (53.6%)	1.22 (0.85–1.74)	0.27	1.21 (0.84–1.74)	0.37

## Data Availability

The data presented in this study are available on request from the corresponding author. The data are not publicly available due to the study protocol’s standard operating procedures on data management and storage, in accordance with the Foundation’s and NHSRC’s guidelines.

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
