# Peer review of "Safety of Janssen Ad26.COV.S and Astra Zeneca AZD1222 COVID-19 Vaccines among Mobile Phone Users in Malawi: Findings from a National Mobile-Based Syndromic Surveillance Survey, July 2021 to December 2021"

_ijerph, 2023, doi:10.3390/ijerph20237123_

Round 1
Reviewer 1 Report
Comments and Suggestions for Authors
COVID-19 should be added to the article title. Readers may not recognize the topic, nor these vaccines, by reliance on reference to clinical trial numbers. It would also be helpful to include the manufacturers s of these vaccines, i.e., Janssen Ad26.COV2.S and Astra Zeneca AZD1222
Nice approach to use random digit dialing from a master list of mobile phone numbers. This is an unusual feature, from my experience. Back translation of translated questionnaire is good practice. That said, It would be helpful to understand how the questions about AEs were asked and if a time frame (time from vax to questionnaire) was included as part of the prompt.
It is unclear what the research assistants did. On line 99, page 3 it says that they were trained….to conduct the interviews. Later on the same page, line 112, the authors mention using tablet with ODK software to capture responses electronically. Which was it?
I would suggest that Sample Size calculation should be focused on the purpose of this study, not general syndromic surveillance, and I would recommend specifying the expected rate on which this study size was calculated.
There were some interesting anomalies in this report. The reported rates of SAEs were high at 2.6%, of note, higher than similar studies in the US for the Janssen vaccine that was used here. Yet only 49% reported at least one AE, which seems low. In the US, 79% reported injection site reactions with only 24.4% here). This merits discussion. I would suggest the authors considering incorporating the time from vaccine date to interview date in their multivariate model.
The discussion section would benefit from a consideration about sampling. Here we learn that only 7% of the Malawi population actually were vaccinated yet the rate in this survey was 36% so there was clearly some selection bias. It might also help to add to the flow chart to show how many people were contacted between July 2021 to Dec 2021and then among those, how many agreed to be interviewed and how many refused.
Some of the most interesting findings from this study are the relation between thinking that COVID-19 vaccines were unsafe and the likelihood of reporting more AE. Not surprising but useful to have this proof point.
Table 4 reports factors associated with reported AE using logistic regression. Glad to see the confidence limits. It’s not clear to me about the information value of pooling all AEs for this analysis. I would think that from a public health perspective it would be more helpful to examine moderate to severe AEs, since people usually aren’t surprised when they feel some pain after any type of injection.
The p-values in table 2 look off and need more information. Some of the distributions look pretty balanced yet showing a p-value makes them look significant, e.g., region. It’s hard to understand the purpose of calculating p-values for several of these descriptive variables, e.g., type of vaccine
Author Response
Response to Reviewer 1 Comments
Point 1: COVID-19 should be added to the article title. Readers may not recognize the topic, nor these vaccines, by reliance on reference to clinical trial numbers. It would also be helpful to include the manufacturers s of these vaccines, i.e., Janssen Ad26.COV2.S and Astra Zeneca AZD1222
Response 1: Noted with thanks. We have updated the title as recommended
Point 2: Nice approach to use random digit dialing from a master list of mobile phone numbers. This is an unusual feature, from my experience. Back translation of translated questionnaire is good practice. That said, It would be helpful to understand how the questions about AEs were asked and if a time frame (time from vax to questionnaire) was included as part of the prompt..
Response 2: Noted with thanks. We have included information on how the questions about the duration between the last vaccine dose and the interview date, and those about the AEs were asked.
Point 3: It is unclear what the research assistants did. On line 99, page 3 it says that they were trained….to conduct the interviews. Later on the same page, line 112, the authors mention using tablet with ODK software to capture responses electronically. Which was it?
Response 3: Noted with thanks.The trained research assistants conducted the interviews and entered the data in real-time in the tablet with ODK software. We have included this statement in the methods section.
Point 4: I would suggest that Sample Size calculation should be focused on the purpose of this study, not general syndromic surveillance, and I would recommend specifying the expected rate on which this study size was calculated.
Response 4: Thank you for your comment. We did not calculate a specific sample size for the vaccine study. However, the sample size, estimated for the original survey's primary objective was considered sufficient for the vaccination study, for a precision estimate of +/- 5% for a commonly reported side effect of pain at the injection site.
Point 5: There were some interesting anomalies in this report. The reported rates of SAEs were high at 2.6%, of note, higher than similar studies in the US for the Janssen vaccine that was used here. Yet only 49% reported at least one AE, which seems low. In the US, 79% reported injection site reactions with only 24.4% here). This merits discussion. I would suggest the authors considering incorporating the time from vaccine date to interview date in their multivariate model.
Response 5: Thank you for the observation. The scope of our study could only assess the severity of AEs as graded by the respondents (2.6% of which were severe), as opposed to the seriousness of the AE (defined as an event that meets one of the following criteria: results in death, is life-threatening,
requires inpatient hospitalization or prolongation of existing hospitalization, results in persistent or significant disability/incapacity, is a congenital anomaly/birth defect), which have been reported at a much lower rate. However, we could assess the rate of hospitalizations (0.7%) which was similar to findings from other studies. The rate of reported injection site reactions was comparable to study in South Africa, and closer to the ranges reported by the CDC (33.3% - 53.8%).
Point 6: The discussion section would benefit from a consideration about sampling. Here we learn that only 7% of the Malawi population actually were vaccinated yet the rate in this survey was 36% so there was clearly some selection bias. It might also help to add to the flow chart to show how many people were contacted between July 2021 to Dec 2021 and then among those, how many agreed to be interviewed and how many refused.
Response 6: Thank you for the recommendation. We included a note in the limitations section contrasting our study participants with the country’s population. We have also added the enrolment cascade between July 2021 and December 2021.
Point 7: Some of the most interesting findings from this study are the relation between thinking that COVID-19 vaccines were unsafe and the likelihood of reporting more AE. Not surprising but useful to have this proof point.
Response 7: Noted with thanks
Point 8: Table 4 reports factors associated with reported AE using logistic regression. Glad to see the confidence limits. It’s not clear to me about the information value of pooling all AEs for this analysis. I would think that from a public health perspective it would be more helpful to examine moderate to severe AEs, since people usually aren’t surprised when they feel some pain after any type of injection.
Response 8: Thank you for the recommendation. We have run another regression model for moderate and severe AEs and are getting a similar picture.
Point 9: The p-values in table 2 look off and need more information. Some of the distributions look pretty balanced yet showing a p-value makes them look significant, e.g., region. It’s hard to understand the purpose of calculating p-values for several of these descriptive variables, e.g., type of vaccine
Response 9: Thank you for the comment. We wanted to demonstrate there were differences beyond chance in the self-report of post-vaccination AE among different sociodemographic groups. While the proportions look similar, the p-values are affected by the large numbers i.e., reaching statistical significance.

Reviewer 2 Report
Comments and Suggestions for Authors
This study, with its relatively simple design but very large sample size, is fairly well described and understandable. Certain methodological points need to be developed further.
My comments can be found directly in the PDF

Author Response
Response to Reviewer 2 Comments
Point 1: you can't draw the same conclusion when the OR's don't all point in the same direction
Response 1: Thank you for the comment. We have revised the results to include the groups with OR’s pointing in the same direction.
Point 2: Questionnaire been tested before on a pilot sample? more information is needed on the choice of questions/variables collected.
Response 2: Thank you for the query. We have updated the methods section to include the piloting process
Point 3: were the interviewers trained beforehand?
Response 3: Thank you for the question. The interviewers were trained in conducting phone-based interviews, the study protocol, and had research ethics certification.
Point 4: in the multivariate model, indicate which variables have been selected for the OR adjustment
Response 4: Thank you for the recommendation. We included the information in the methods section.
Point 5: if you have the 95% CI, you don't need the p-value.
Response 5: Noted with thanks. However, we prefer to present the p-values as well. Please see:
du Prel JB, Hommel G, Röhrig B, Blettner M. Confidence interval or p-value?: part 4 of a series on evaluation of scientific publications. Dtsch Arztebl Int. 2009 May;106(19):335-9. doi: 10.3238/arztebl.2009.0335. Epub 2009 May 8. PMID: 19547734; PMCID: PMC2689604.
Additionally:
P-values and confidence intervals are two different ways of assessing the evidence provided by a statistical test. P-values are commonly used to determine whether a finding is statistically significant, while confidence intervals are used to estimate the range of values that are likely to contain the true population parameter. Here are some benefits of using both:
- P-values can give a clear cut-off for statistical significance, which can be useful for making decisions about whether to accept or reject a null hypothesis.
- Confidence intervals give a range of plausible values for the population parameter, which can be more informative than a single point estimate.
- Confidence intervals can help to communicate the uncertainty associated with a particular estimate, which can be helpful for interpreting the results of a study.
- In some cases, a p-value and a confidence interval can give different results, which can help to identify potential issues with the data or the analysis.
It's also worth noting that p-values and confidence intervals are not mutually exclusive and can be used together to provide a more complete picture of the evidence provided by a statistical test.
